# Towards Intelligent, Programmable, and Open Railway Networks

Evelina Pencheva [1,*], Ivaylo Atanasov [2] and Ventsislav Trifonov [2]

1 Telecommunications Department, Faculty of Telecommunications and Electrical Equipment in Transport, "Todor Kableshkov" University of Transport, 1574 Sofia, Bulgaria

2 Communication Networks Department, Faculty of Telecommunications, Technical University of Sofia, 1756 Sofia, Bulgaria; iia@tu-sofia.bg (I.A.); vgt@tu-sofia.bg (V.T.)

\* Correspondence: evelina.nik.pencheva@gmail.com

**Abstract:** The virtualization and automation of network functions will be key features of future high-speed railway networks, which have to provide dependable, safe, and secure services. The virtualization of railway network functions will enable functions such as train control, train integrity protection, shunting control, and trackside monitoring and maintenance to be virtualized and to be run on general-purpose hardware. Network function virtualization combined with edge computing can deliver dynamic, low-latency, and reliable services. The automation of railway operations can be achieved by embedding intelligence into the network to optimize the railway operation performance and to enhance the passenger experience. This paper presents an innovative railway network architecture that features distributed intelligence, function cloudification and virtualization, openness, and programmability. The focus is on time-tolerant and time-sensitive intelligent services designed to follow the principles of service-oriented architecture. The interaction between identified logical identities is illustrated by use cases. The paper provides some details of the design of the interface between distributed intelligent services and presents the results of an emulation of the interface performance.

**Keywords:** European Train Control System; artificial intelligence; network function virtualization; service-oriented architecture; concurrent process modeling; key performance indicators



## 1. Introduction

The digitalization and embedding of innovation technologies are expected to enhance the capacity and optimize the performance of railway operations and to improve the quality of experience for passengers [1]. At the same time, the environmental requirements of society for transport operations have significantly increased, and the criteria for mechanical and acoustic pollution of the environment and carbon footprints have become crucial. To protect the environment and to address the ever-increasing demands for high-efficiency railway transport, all processes related to train control, train protection and supervision, trackside monitoring, and passenger services need to be automated. The automation of data-exchange processes reduces the possibility of human errors, decreases delays or bottlenecks in emergencies, streamlines data transfer, conserves resources, and increases automated reporting transparency and control [2,3].

The European Railway Traffic Management System (ERTMS) was designed to provide a highly reliable, fail-safe, and interoperable European solution for proper operation of railways and an enhanced passenger experience [4]. The project was developed to deploy a single control, command, and signaling system [5]. Soon, the ERTMS gained popularity and was deployed not only in Europe, but also in Asia, Africa, and South America as one of the pillars of transport decarbonization policy. The ERTMS is composed of the European Train Control System (ETCS) and the Global System for Mobile Communications on Railways (GSM-R). The ETCS consists of onboard equipment for supervising the train's movements and trackside-deployed balises that store data on the railway's infrastructure. The GSM-R

is intended to provide mobile communications between trains, trackside equipment, and a traffic control center. The interfaces installed onboard the train and the trackside system exchange information to control and operate at the maximum allowed train speed.

The ERTMS/ETCS can be configured to operate on different levels related to the trackside equipment deployed, the way the information from the trackside is presented to the onboard units, and the distribution of functions between the onboard and trackside equipment. The radio block center (RBC) is an essential ETCS component on Level 2 and Level 3. It is responsible for train guidance and monitoring by receiving the necessary information from the train and trackside equipment and by generating the ETCS travel permits that are sent via the communication network [6,7].

The Future Railway Mobile Communication System (FRMSC) is the successor of the GSM-R, with the objective of becoming a worldwide standard [8,9]. The work on the standardization of the FRMCS started with identifying user requirements and went through defining typical use cases and describing the principle FRMCS architecture. The FRMCS will use the public telecommunication networks to connect onboard users, such as automatic train control (ATC)/automatic train protection (ATP), train drivers, shunting staff, passengers, security staff, sensors, and SOS stations, with ground users in data centers for ATC/ATP, in railway offices, at stations and platforms, at depots, and on the trackside. Current fifth-generation (5G) communications can address the requirements of the FRMCS by offering enhanced broadband service, ultrareliable and low-latency services, and massive machine-type services [10–12].

Fifth-generation (5G) communications and artificial intelligence (AI) are two essential ingredients that drive future innovations in the railway industry [13,14]. 5G communications make it possible to provide connectivity, and AI can provide the ability of onboard, trackside, and ground devices to not only perceive, reason, and act intuitively, but also to solve technical challenges. AI advancements can help to improve railways' performance and efficiency, while 5G connectivity of railway assets can fuel distributed intelligence at the edge. Edge intelligence can satisfy the rising requirements of critical railway applications [15–17].

The authors' previous research presented the idea of the disaggregation of the ETCS's radio block center functionality following the principles of openness and intelligence [18]. In this paper, the authors elaborate the idea for a cloud-based, virtualized, programmable, and disaggregated railway network architecture. The separation of control functions from the hardware equipment and the usage of standardized control interfaces have the potential to empower the definition of customized closed control loops that will embed real-time analytics and intelligence in railway operations and, thus, pave the way towards autonomation.

The rest of the paper is organized as follows. The next section provides a brief description of the ETCS levels and the ERTMS/ETCS system architecture, with a focus on the radio block center as a main control point in ETCS Levels 2 and 3. Section 3 gives an overview of AI use cases in the railway sector. Sections 4 and 5 discuss how the intelligence may be distributed in the ETCS system architecture, thus making it programmable, open, and virtualized, and then present some issues related to the modeling of distributed intelligence. Section 6 discusses the results from an emulation of the interface between distributed intelligent control functions. The conclusions summarize the authors' contributions.

## 2. ETCS Levels and Architecture

The deployment of ETCS application levels depends on the existing railway infrastructure.

ETCS Level 1 provides train cab signaling where movement authorities can be provided through switchable, fixed Eurobalises and Euroloops. The Eurobalises installed on the trackside send data about the train route to the onboard equipment, and based on the data, the maximum train speed and breaking curves are calculated. In addition, the Euroloops (loop infill) or radio-infill solutions are used for transmission of distant signal data to trains over a long distance.

ETCS Level 2, the radio block center (RBC), sends movement authorities to the onboard equipment using the GSM-R, so no line-side signals are required. Eurobalises, acting as

passive positioning beacons, help the train to calculate its position, which is refined by additional sensors.

In ETCS Level 3, the trains continuously monitor their own position and provide train position data to the RBC. Level 3 is a fully radio-based system without track-based detection equipment. Based on the received train positioning data, the RBC calculates the smallest safe train distance at any time. The role of the train integrity functions is crucial as the trackside equipment disappears. ETCS Level 3 is under standardization.

The ERTMS/ETCS system architecture includes onboard ETCS equipment and trackside ETCS equipment [19,20]. The onboard equipment consists of a central logic unit, rail path sensors, control devices, cab displays, and a module for radio communications. The trackside equipment is a fixed part of the ETCS installation, and depending on the ETCS level, it can be composed of Eurobalises, Euroloops, a lineside electronic unit (LEU), radio infill unit (RIU), key management center (KMC), interlocking system, and RBC.

AnEurobalise is a system for the transmission of safety-relevant information between the train and the trackside equipment. It consists of beacons situated along the tracks that transmit when a train antenna is above them, an onboard balise transmission module and antenna unit, and a trackside signaling system. Fixed data balises provide special information, such as speed restrictions and gradients. Transparent data balises are connected to the LEU, which transmits dynamic data to the train, e.g., signal indications. An Euroloop is a leaky feeder system that transmits the signal aspects to the trains. An RIU is a radio-based communication system that allows to send the message of the next signal in the travel direction in advance to the train before passing the relevant information point. The interlocking system is responsible for the supervision and safety control of routes, switch points, signals, and track locks. It prevents conflicting train movements through arrangements of tracks, such as crossings and junctions. The KMC is responsible for the management (installation, update, and deletion) of the cryptographic keys that are used to secure radio communications between the ETCS entities.

The main task of the RBC is to monitor and guide the trains in its area. It uses the train and trackside information to generate movement authorities and to send them to the trains by radio. The implementation of the RBC function depends on the existing railway infrastructure and the operator's requirements. The functions, architecture, and user interfaces of the RBC are not standardized, are structured differently by various manufacturers, and are centered around safety-relevant functionality. Typical RBC functions include generating movement authorities, monitoring trains' movements in the RBC area, announcing radio gaps, evaluating potential hazardous situations, triggering emergency stops, the first assignment of a train route, joining and splitting of trains, adaptive speed control, stop evaluation, sending the national values to trains, determining safe reactions in case of dangerous situations on a train, etc.

The operation of the RBC essentially enables dispatcher control and maintenance access. The dispatcher control enables the input, query, and activation of temporary data, such as changes in ETCS modes, emergency breaks, speed restrictions, etc. The access to maintenance provides access to and storage of log files, retrieval of information for malfunctions, and RBC software configurations/updates/restart. The RBC's user interfaces make train tracking possible. Depending on the existing trackside infrastructure, the operation of the RBC may be partially or fully automated. The tendency is towards the full automation of the RBC operation based on embedded intelligent decisions.

The specifications do not define any details on the implementation of RBC, just the operating modes and scenarios, and based on operational processes, the railway operator determines the RBC requirements. So, the current RBC implementations are proprietary and monolithic solutions.

A possible future evolution of the RBC architecture may be related to disaggregation. Horizontal disaggregation can be achieved through the separation of time-tolerant RBC functions from RBC time-sensitive RBC functions using open interfaces. The separation of the RBC's functionality aims at running different control loops with distinct latency require-

ments. Interface openness can be achieved through application programming interfaces (APIs) that facilitate interoperability. The valuable benefits of APIs are related to a low cost and easy implementation, more flexible functionality, cloud-based deployment, etc. Vertical disaggregation decouples hardware and software by applying the virtualization of RBC functions. Function virtualization can be achieved by transitioning from software running on proprietary and specialized hardware to software running on general-purpose hardware. Among the advantages of function virtualization are the reduction of maintenance costs, easier function upgrades, scalability, greater agility, and flexibility.

The disaggregation of software and data channels by introducing AI/machine learning (ML) techniques and connectivity to external contextual data sources can contribute to the optimization of railway operations.

Currently, the AI technologies that are most often used on railways include ML, robotics, and natural language processing [21]. In the rest of the paper, the focus is on ML management in railway transport when used to perform tasks or to make predictions based on experience or example data. In this context, ML can be used for sensing and prediction. For example, sensing can be useful in image recognition for dangerous situations and image analysis for obstacle detection, fault detection, and maintenance of tunnels, bridges, buildings, etc. Prediction is useful in, e.g., preventive maintenance of the equipment, and it can improve railway safety, punctuality, reliability, and availability.

The next section presents a brief description of the research related to the implementation of AI with a stress on ML in the railway sector.

## 3. AI Technologies within the Railway Sector

Use cases of the deployment of AI technologies in railways and perspectives thereof are described in [21]. ML can be effective in image recognition in the fight against terrorism. Data collection and processing using AI algorithms may help to predict abnormal behavior in passengers [22,23]. ML can be used for predictive maintenance, e.g., on rolling stocks, infrastructure, and other operational decision support. Rolling stock maintenance is important for railway service availability, punctuality, and reliability, and ML algorithms may predict wheel tread defects or recognize anomalies in train bearing temperature, etc. [24]. Train punctuality and safety, among the other things, depend on the reliability and availability of railway infrastructure, including tunnels, bridges, tracks, cuttings, and embankments [25,26]. ML can be used in the optimization of traffic management, train path allocation, passenger flow management at railway stations, detection of abandoned luggage, etc. [27].

In addition to the application areas listed above, AI/ML techniques may be used for autonomous train control and train driving, transferring the responsibilities from manual operators to an onboard control system [28]. A review on the application of AI in high-speed railway automatic train control is provided in [29]. The existing research is mainly focused on speed control, trajectory control, and intelligent monitoring and prediction of abnormal conditions.

Some of the RBC functions are time sensitive, while others are delay tolerant. Actions related to ATC and ATP, as well as some measures related to preventive maintenance based on ML models, must be implemented in near real time, while others are non-real time. The proposed distributed intelligent railway network architecture disaggregates the RBC's functionality into time-sensitive control functions and time-insensitive control functions.

The idea for the disaggregation of the RBC's functionality is inspired by the concept of an open radio access network (O-RAN) that is aimed at the provisioning of an open multi-vendor platform and embedding ML to optimize the RAN's performance [30].

The next section provides more details on the proposed disaggregated railway network architecture, which encourages programmability, openness, virtualization, and intelligence.

## 4. Intelligent Railway Network Architecture

Figure 1 shows a high-level view of the proposed open, intelligent, and programmable railway network architecture, with a focus on the separation of time-tolerant RBC functions from time-sensitive RBC functions. Radio communication modules for information exchange based on the FRMCS are not shown for simplicity.

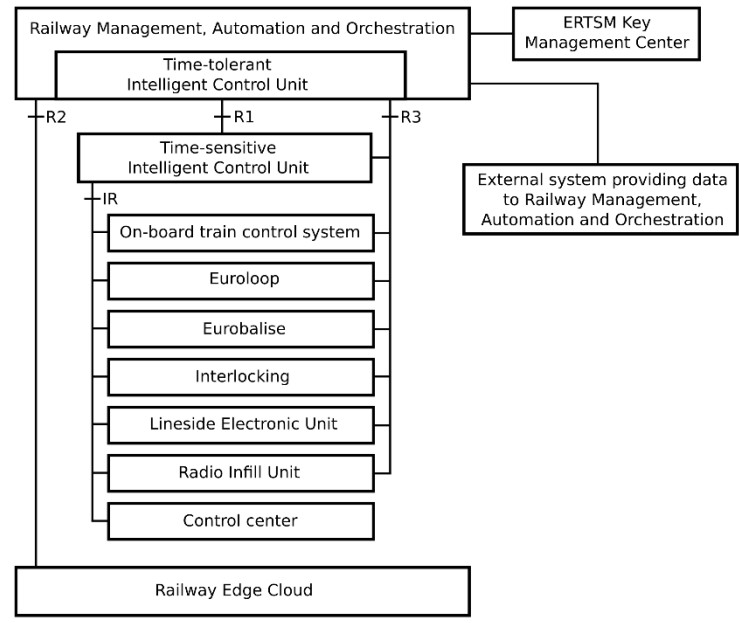

**Figure 1.** High-level view of the logical architecture of the intelligent railway network.

The vertical disaggregation can be achieved through the virtualization of RBC functions. The virtualization enables software to run on a standard server platform and provides greater scalability, adaptability, and agility compared to traditional proprietary RBC equipment. The virtualization of the RBC functions can provide an open architecture with flexible deployment options. Virtualized RBC functions can be implemented as software applications that deliver specific RBC functions, such as train monitoring and control, track-side monitoring and management, security, radio communications, etc. These software applications can be packaged as virtual machines or containers that run on the railway edge cloud (REC), which is deployed at the edge of the railway network.

The REC is a cloud computing platform that consists of computing, storage, and networking components that meet the requirements to host the relevant RBC functions, supporting software components such as a hypervisor, container runtime, etc., as well as appropriate management and orchestration functions. The REC exports interfaces for cloud and workload management (e.g., infrastructure discovery, software and workload lifecycle management, and registration).

The railway management, automation, and orchestration (RMAO) framework is responsible for the management and orchestration of virtualized RBC functions. It supports fault, configuration, performance, and software management and initial installation over the R3 interface. The RMAO hosts a time-tolerant intelligent control unit (TTICU) whose primary goal is to support intelligent railway operation optimization by providing ML model management, policy-based guidance, and supplementary information to the time-sensitive intelligent control unit (TSICU).

The TTICU performs intelligent, non-real-time RBC functions, such as the first route assignment for the train, train chasing, acquisition of diagnostic information, trackside maintenance, etc. The TTICU can embed intelligence to optimize railway operation, for which it may use RMAO services for data collection and provisioning services for an onboard train control system, an interlocking system, and a control center. The TTICU contains a framework functionality, which terminates the interface towards the TSICU and

exposes the required services to ttApps. The ttApps are modular applications used for railway operation optimization. The TTICU is involved in a non-real-time control loop.

The TSICU's functionality includes time-sensitive monitoring, control, and optimization of the onboard train control and trackside equipment functions. It is involved in a near-real-time control loop (e.g., responsible for setting temporary train speed limits, evaluation of potentially dangerous situations, commanding emergency stops, etc.). It hosts tsApps that use the interfaces toward the ETCS entities to collect time-sensitive information (e.g., on a train basis or a track-section basis) and provide value-added services. The TSICU control steers the ETCS entities (onboard train control system, interlockings, LEU, RIU, Eurobalise, Euroloop) through policies and enrichment information provided by the TTICU.

The R1 interface is between the TTICU in the RMAO and the TSICU for railway operation optimization. It enables policy management, information provisioning, and ML model deployment and updating. It is used to provide the TSICU with enrichment information, policies, and ML model updates, as well as to provide feedback information to the TTICU on how the policy set works. The R1 application protocol is based on the REST (REpresentational State Transfer) solution and uses hypertext transfer protocol (HTTP) procedures and JavaScript object notation (JSON) objects. The R1 policies are not critical to train control and trackside control, have temporary validity, and may be dedicated for an individual onboard train control system, Euroloop, interlockings, or dynamically defined groups of ETCS entities.

The R2 interface between the RMAO and the REC provides platform resources and workload management. It enables the deployment and life-cycle management of virtualized functions that run on the REC.

The R3 interface is between the RMAO and the railway network functions for management support. It is used for functions by which software management, file management, configuration management, and fault management may be achieved.

The IR interfaces connect the onboard train control units, Euroloops, interlockings, and control center to the TSICU. The procedures executed over the IR interface are related to the exchange of information between ETCS entities or to notifications about internal events or events triggered by external entities and the required reaction. These procedures define the required status and mode changes of the ETCS entities (train driver, the trackside equipment, the onboard equipment) on a contextual level. Examples of procedures include mission starting and mission ending, shunting, and overriding, e.g., in case of a failed signal, train joining or splitting, train orientation changing, and indicating of track conditions. The full list of procedures and their detailed descriptions may be found in [31].

The proposed intelligent railway network architecture supports at least three control loops involving different RBC functions:

- Non-real-time control loops running at the TTICU level;
- Near-real-time control loops running at the TSICU level;
- Real-time control loops running at the level of the onboard train control system and trackside equipment.

The control loops are executed simultaneously at different levels and may or may not interact with each other depending on the use case. The typical execution time for use cases involving non-real-time control loops is 1 or more seconds; the near-real-time control loops are on the order of 10 milliseconds or more; control loops in the ETCS entities can operate below 10 milliseconds. The timing of the control loops is dependent on the use case.

The following use case illustrates the role of the logical entities in the proposed intelligent railway architecture. The use case describes the motivation and solution for temporary train speed control based on inspection of railway tracks and preventive maintenance. One of the top causes of train derailing is track-related issues. ML-based trackside equipment maintenance can predict fault occurrence before assets fail and can issue train speed commands in the affected area to avoid taking a critical train out of service and disrupting the system.

Numerous datasets covering different domains of the railway sector and that have been used to provide ML-based solutions are reviewed in [32]. One of the ML-based solutions described gathers information about the mechanical behavior, the railway track state, and reasons for possible problems. A sensor system using optical fibers monitors tracks and collects information about track performance. The railway track performance may vary significantly along the track length, and knowledge about it helps in track maintenance; in some cases, it may be critical for train speed along the track.

The goal of the use case is to determine the temporary train speed along a track section based on knowledge about track performance. The entities involved are as follows. The TTICU is responsible for training, deployment, and updating of ML models related to track performance in the TSICU, and for rail policy generation and provisioning. The TSICU executes the deployed ML models, converts the policy related to the temporary train speed into train commands, and sends train speed and track performance data to the TTICU for evaluation and optimization. The onboard train control system enforces the policy by following the commands sent by the TSICU. The trackside equipment provides measurement data, and the RMAO performs data collection and control as a termination point of the R3 interface.

Figure 2 illustrates the procedure of the training and deployment of ML models. As preconditions, the collection and control functionalities in the RMAO have an established data collection and sharing process, and the TTICU has access to these data. The TTICU monitors track performance by collecting the relevant track performance events.

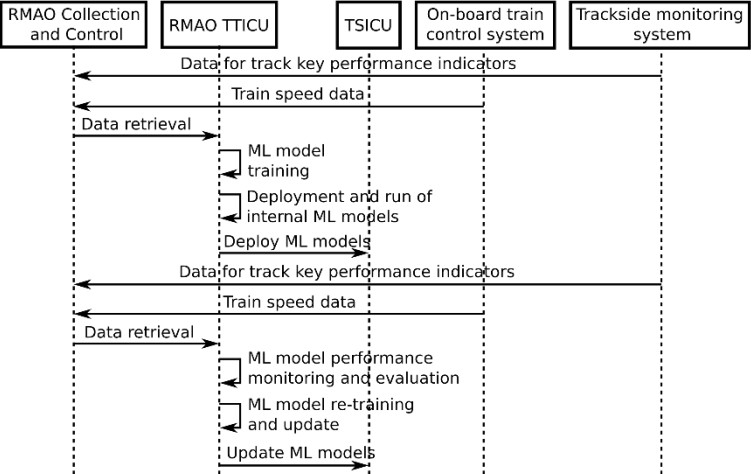

**Figure 2.** Training, deployment, and updating of the ML model for track performance.

The railway operator has set a trigger condition related to the monitored track parameters.

1. Upon detection of the triggered event, the TTICU accesses the track performance and train speed data from the RMAO and decides either to create a new ML model for track performance or to modify an existing one.
2. The TTICU performs ML model training, obtains track-performance-related models, and may internally deploy a train speed model.
3. The TTICU deploys or updates the ML model in the TSICU via the R1 interface.
4. The TSICU stores the received ML model.
5. The TTICU may configure, if required, a specific track performance measurement to collect the data required to assess the performance of the deployed ML model. Based on an evaluation of the ML model's performance and model retraining, the TTICU may update the ML model in the TSICU.
6. The procedure ends when a trigger condition specified by the railway operator is satisfied.

As a post-condition, the TSICU stores the received track performance ML model and train speed model and executes the models for the dynamic optimization of the train speed control function.

Figure 3 illustrates the procedure for rail policy generation, policy enforcement, and policy performance evaluation. As a precondition, track-performance-related models and train speed models have been deployed in the TTICU and TSICU.

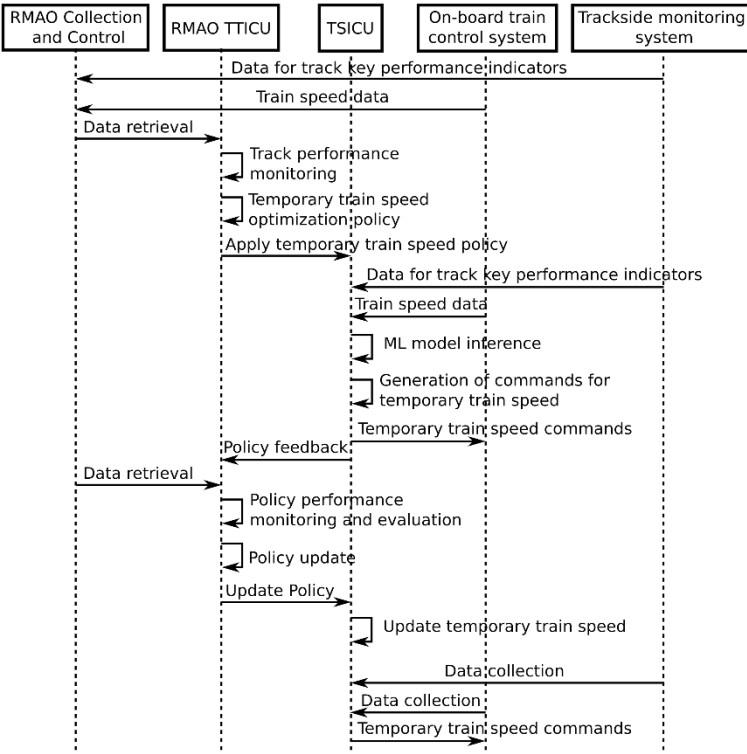

**Figure 3.** Train speed policy generation, policy enforcement, and policy performance evaluation.

1. The railway operator wants to generate a temporary train speed policy or to optimize the ML models.
2. The TTICU sends a train speed policy to the TSICU via the R1 interface.
3. The TSICU receives the train speed policy, infers about the track performance and train speed based on the ML models, and translates the policy into train speed control commands.
4. The TSICU sends the train speed control commands to the onboard train control system to optimize the train speed.
5. The onboard train control system enforces the commands received from the TSICU.
6. The TTICU may optionally receive policy feedback regarding the train speed from the TSICU, data from track performance measurements, and performance data of the train speed optimization function in the TSICU, and it may update the train speed policy.
7. The procedure ends when a trigger condition specified by the railway operator is satisfied.

As a post-condition, the TTICU collects and monitors the relevant key performance indicators from the onboard train control system to observe the performance of the train speed optimization function in the TSICU.

The research focus is on the R1 interface, so the next section provides more details on its description.

## 5. Interface between Time-Tolerant and Time-Sensitive Functions

The TTICU, which resides in the RMAO framework, can be connected to multiple TSICUs, which, in turn, can control trains, Euroloops, interlockings, and RIUs in their areas, as shown in Figure 4. As an alternative, the TSICU may be implemented in one physical node consisting of more logical TSICUs with specialized functions for train control and trackside equipment control.

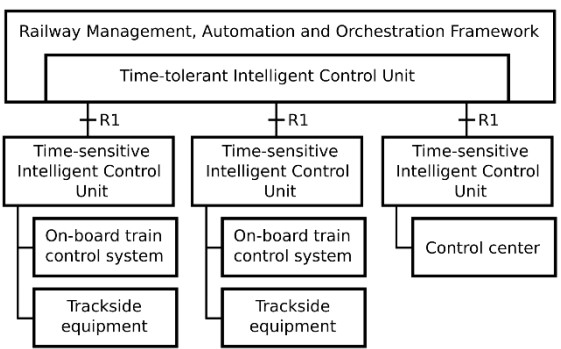

**Figure 4.** Deployment of time-tolerant and time-sensitive intelligence.

The purpose of the R1 interface is to enable the TTICU functions to provide policy-based guidance, ML model management, and enrichment information to the TSICU functions so that the railway operation be optimized under certain conditions.

Based on observables such as events and trigger conditions provided via the R3 interface, the TTICU may define policies that are provided to the TSICU via the R1 interface. The TSICU enforces the policy and collects train and trackside equipment data via the R3 interface and provides the policy feedback to the TTICU via the R1 interface. The TTICU uses this information to continuously evaluate the impact of the R1 policies towards the fulfillment of railway intent. Based on internal conditions, the TTICU may decide to issue new policies or to update existing policies.

The ML models can be trained and executed at different places in the intelligent railway network architecture.

In one scenario, the ML models may be trained on the RMAO layer and then used by the TTICU to improve the performance monitoring and guidance of the train and trackside equipment based on the R3 observability. The training and deployment of the ML model may be handled by the internal RMAO functions and the same training data to be used for TTICU inference.

In another scenario, the ML models may be trained in the RMAO layer and then used by the TTICU to optimize the train and trackside equipment performance. The training and deployment of the ML model may be handled by the internal RMAO functions, and the training data used for the TSICU inference are the same. The trained ML models may be deployed by the RMAO layer via the R3 interface. The performance evaluation of the ML model and the handling of explicit feedback from the ML model itself are based on the R3 observability.

The TSICU's functions are based on its internal functionality or tsApps, the configuration received via R3, and the R1 policies. The R1 enrichment information service can be used to support the policy enforcement in the TSICU. The enrichment information, which is provided in addition to the generally available information, may enhance railway operation performance and may be gathered by the RMAO from different internal and external sources. The TTICU provides enrichment information to the TSICU via the R1 interface.

Following the service-oriented approach, the time-tolerant intelligent services and time-sensitive intelligent services may be defined according to the REST principles. REST is an architectural style for distributed applications that recognizes everything as a resource. The resource is uniquely identified, supports a standard interface, and has a JSON representation that is exchanged across the network over hypertext transfer protocol secure (HTTPS). The protocol stack of the R1 interface is shown in Figure 5.

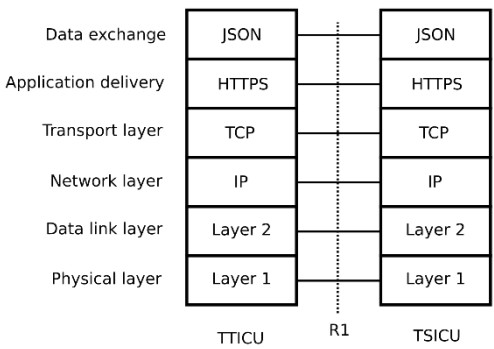

**Figure 5.** Protocol stack of the R1 interface.

In REST, any resources can be created (HTTP POST method), updated (HTTP PUT method), deleted (HTTP DELETE method), and read (HTTP GET method).

Figure 6 shows the uniform resource identifier (URI) structure of the railway policy resources.

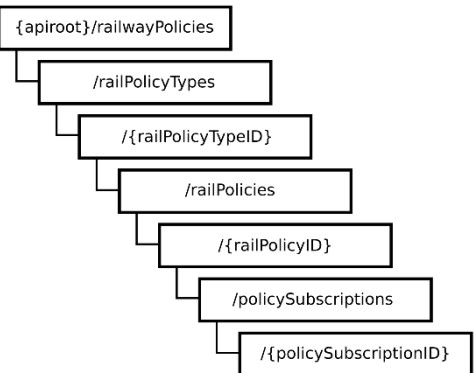

**Figure 6.** Structure of URI of the railway policy resources.

The railPolicyTypes resource is a container of all policy types. It supports an HTTP GET method that retrieves a list of rail policy identifiers. The railPolicyTypeID resource represents an individual rail policy type, and an HTTP GET method applied on the resource retrieves the respective policy type information. The railPolicies resource represents all rail policies from a certain type, and the list of policies may be retrieved using an HTTP GET method, while an HTTP POST method creates a new rail policy. The railPolicyID method represents an individual rail policy. The HTTP methods supported by the railPolicyID resource include PUT, which updates an existing rail policy, GET, which retrieves information about the rail policy, and DELETE, which is used to remove the rail policy.

Figure 7 shows the flow of the rail policy creation procedure. The TTICU generates the rail policy identifier and sends an HTTP POST request to the TSICU. The target URI identifies the resource under which a new rail policy has to be created. The message body carries the JSON policy description. On success, the TSICU returns a "201 Created" response. On failure, the respective error code is returned.

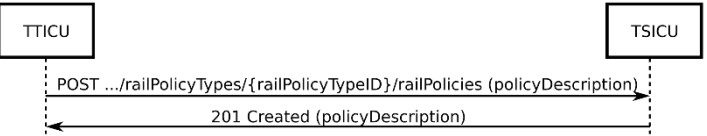

**Figure 7.** Flow of the rail policy creation procedure.

Figure 8 shows the flow to query a single rail policy. The TTICU queries about an individual rail policy by sending an HTTP GET method with a URI pointing to the resource representing the rail policy of interest. The response returns the policy description and status.

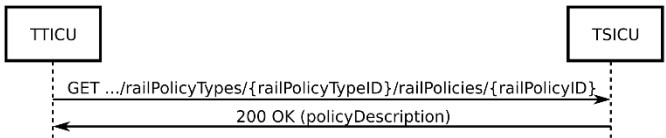

**Figure 8.** Flow of the rail policy query procedure.

To provide policy feedback, subscriptions to rail policies have to be supported. The policySubscriptions resource represents all subscriptions for notifications related to an individual rail policy. The list of all policy subscriptions may be retrieved by the HTTP GET method. The TTICU creates a new subscription by sending an HTTP POST method with a message body describing the criteria for notifications and the callback address at which it wants to receive notifications. The policySubscriptionID resource represents an individual subscription for a specific rail policy. It supports the HTTP GET method, which is used to retrieve subscription information, the HTTP PUT method, which is used to modify the subscription, and the HTTP DELETE method, which is used to terminate the subscription.

Figure 9 shows the flow of the subscription creation procedure. Figure 10 shows the flow of the notification procedure, which provides policy feedback to the callback address indicated earlier in the subscription request.

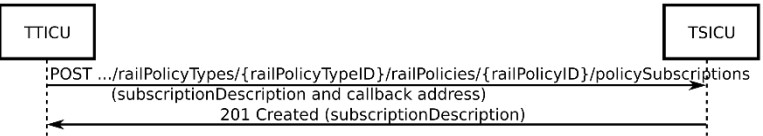

**Figure 9.** Flow of the subscription creation procedure.

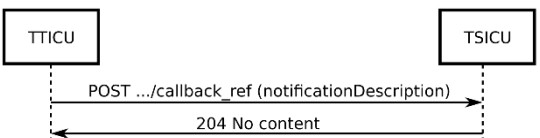

**Figure 10.** Flow of the notification procedure.

The TTICU and TSICU must have synchronized views on the railway policy status.

Figure 11 shows a model representing the railway policy lifecycle as seen by the TTICU, and Figure 12 shows a model representing the policy state supported by the TSICU. Both models are simplified, as they do not describe the behavior related to subscriptions and notifications.

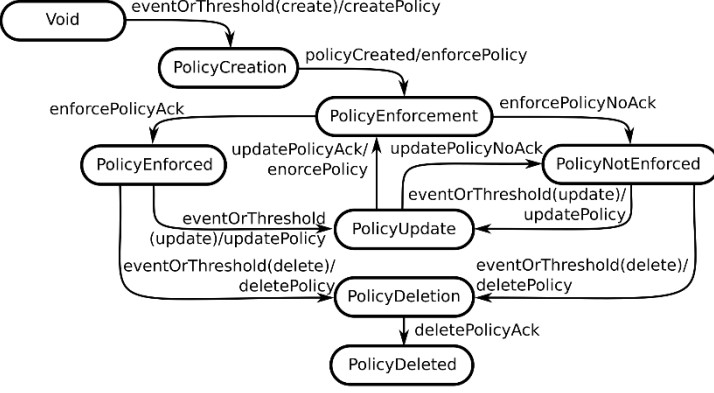

**Figure 11.** Model representing the rail policy lifecycle supported by the TTICU.

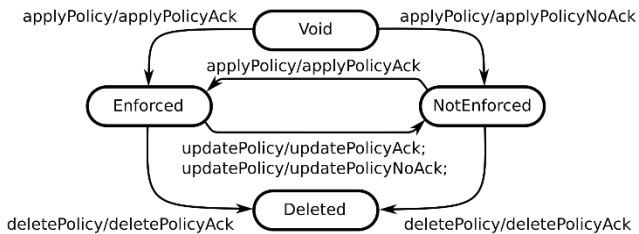

**Figure 12.** Model representing the rail policy state as seen by the TSICU.

From the TTICU's point of view, a policy may be created, updated, or deleted when predefined measurement thresholds are reached. The TTICU can request the enforcement of a newly created or updated policy or a policy deletion. Upon receiving a request from the TTICU, the TSICU converts the policy into corresponding commands, enforces the commands via the IR interface, and sends back the result.

To prove that both state machines expose equivalent behaviors, they are formally described as labeled transition systems (LTSs), and the mathematical formalism of a weak bi-simulation is used. An LTS is defined as a quadruple of a set of states, a set of stimuli that trigger transitions, a set of transitions, and a set of initial states [33,34].

By $T_{TT} = \{S_{TT}, A_{TT}, \rightarrow_{TT}, s_{TT}\}$, an LTS is denoted, which represents the state machine of a policy lifecycle supported by the TTICU, where:

$S_{TT} = \{$Void $[s^{TT}_1]$, PolicyCreation $[s^{TT}_2]$, PolicyEnforcement $[s^{TT}_3]$, PolicyEnforced $[s^{TT}_4]$, PolicyNotEnforced $[s^{TT}_5]$, PolicyUpdate $[s^{TT}_6]$, PolicyDeletion $[s^{TT}_7]$, PolicyDeleted $[s^{TT}_8]\}$.

$A_{TT} = \{$measurementThreshold(create) $[a^{TT}_1]$, policyCreated $[a^{TT}_2]$, applyPolicyAck $[a^{TT}_3]$, applyPolicyNoAck $[a^{TT}_4]$, measurementThreshold(update) $[a^{TT}_5]$, updatePolicyAck $[a^{TT}_6]$, updatePolicyNoAck $[a^{TT}_7]$, measurementThreshold (delete) $[a^{TT}_8]$, deletePolicyAck $[a^{TT}_9]\}$.

$\rightarrow_{TT} = \{(s^{TT}_1\ a^{TT}_1\ s^{TT}_2), (s^{TT}_2\ a^{TT}_2\ s^{TT}_3), (s^{TT}_3\ a^{TT}_3\ s^{TT}_4), (s^{TT}_3\ a^{TT}_4\ s^{TT}_5), (s^{TT}_4\ a^{TT}_5\ s^{TT}_6), (s^{TT}_5\ a^{TT}_5\ s^{TT}_6), (s^{TT}_6\ a^{TT}_7\ s^{TT}_5), (s^{TT}_6\ a^{TT}_6\ s^{TT}_3), (s^{TT}_4\ a^{TT}_8\ s^{TT}_7), (s^{TT}_5\ a^{TT}_8\ s^{TT}_7), (s^{TT}_7\ a^{TT}_9\ s^{TT}_8)\}$.

$s_{TT} = s^{TT}_1$.

Short notations for the names of states, stimuli, and transitions are given in brackets.

By $T_{TS} = \{S_{TS}, A_{TS}, \rightarrow_{TST}, s_{TS}\}$, an LTS is denoted, which represents the state machine of a policy supported by the TSICU, where:

$S_{TS} = \{$Void $[s^{TS}_1]$, Enforced $[s^{TS}_2]$, NotEnforced $[s^{TS}_3]$, Deleted $[s^{TS}_4]\}$.

$A_{TS} = \{$applyPolicy $[a^{TS}_1]$, updatePolicy $[a^{TS}_2]$, deletePolicy $[a^{TS}_3]\}$.

$\rightarrow_{TS} = \{(s^{TS}_1\ a^{TS}_1\ s^{TS}_2), (s^{TS}_1\ a^{TS}_1\ s^{TS}_3), (s^{TS}_2\ a^{TS}_2\ s^{TS}_3), (s^{TS}_3\ a^{TS}_2\ s^{TS}_3), (s^{TS}_3\ a^{TS}_1\ s^{TS}_3), (s^{TS}_3\ a^{TS}_1\ s^{TS}_2), (s^{TS}_2\ a^{TS}_3\ s^{TS}_4), (s^{TS}_3\ a^{TS}_3\ s^{TS}_4)\}$.

$s_{TS} = s^{TS}_1$.

Bi-simulation is a binary relationship between the states of two LTSs that associates the LTSs' behavior as equivalent, i.e., one LTS simulates the other LTS and vice versa. The concept is used to prove the behavioral equivalence of concurrent processes [35,36]. While a strong bi-simulation requires a strict correspondence between the states of the two LTSs, a weak bi-simulation means that there may be internal states and transitions that are not visible to external observers.

**Proposition 1.** $T_{TT}$ and $T_{TS}$ have a weak bi-simulation relationship.

**Proof.** By R, a relationship between the states of $T_{TT}$ and $T_{TS}$ is denoted where:
R = $\{(s^{TT}_1, s^{TS}_1), (s^{TT}_4, s^{TS}_2), (s^{TT}_5, s^{TS}_3), (s^{TT}_8, s^{TS}_4)\}$.
To prove the existence of a weak bi-simulation between $T_{TT}$ and $T_{TS}$, it is necessary to show that all transitions from states in a couple in R terminate into states in a couple of R. The following transition mapping may be identified:

1. The TTICU creates a new policy and requests policy enforcement, and the TSICU responds with a policy enforcement acknowledgement: for $\forall$ $(s^{TT}_1\ a^{TT}_1\ s^{TT}_2)$, $(s^{TT}_2\ a^{TT}_2\ s^{TT}_3)$, $(s^{TT}_3\ a^{TT}_3\ s^{TT}_4)$ $\exists$ $(s^{TS}_1\ a^{TS}_1\ s^{TS}_2)$.

2. The TTICU creates a new policy and requests policy enforcement, and the TSICU does not acknowledge the policy enforcement: for $\forall$ $(s^{TT}_1\ a^{TT}_1\ s^{TT}_2)$, $(s^{TT}_2\ a^{TT}_2\ s^{TT}_3)$, $(s^{TT}_3\ a^{TT}_4\ s^{TT}_5)$ $\exists$ $(s^{TS}_1\ a^{TS}_1\ s^{TS}_3)$.

3. The TTICU updates a policy that has been enforced and requests the enforcement of the updated policy, and the TSICU acknowledges the policy enforcement: for $\forall$ $(s^{TT}_4\ a^{TT}_5\ s^{TT}_6)$, $(s^{TT}_6\ a^{TT}_6\ s^{TT}_3)$, $(s^{TT}_3\ a^{TT}_3\ s^{TT}_4)$ $\exists$ $(s^{TS}_2\ a^{TS}_2\ s^{TS}_3)$, $(s^{TS}_3\ a^{TS}_1\ s^{TS}_2)$.

4. The TTICU updates a policy that has not been enforced and requests the enforcement of the updated policy, and the TSICU acknowledges the policy enforcement: for $\forall$ $(s^{TT}_5\ a^{TT}_5\ s^{TT}_6)$, $(s^{TT}_6\ a^{TT}_6\ s^{TT}_3)$, $(s^{TT}_3\ a^{TT}_3\ s^{TT}_4)$ $\exists$ $(s^{TS}_3\ a^{TS}_2\ s^{TS}_3)$, $(s^{TS}_3\ a^{TS}_1\ s^{TS}_2)$.

5. The TTICU requests an update of a policy that has been enforced and the TSICU does not acknowledge the policy update: for $\forall$ $(s^{TT}_4\ a^{TT}_5\ s^{TT}_6)$, $(s^{TT}_6\ a^{TT}_7\ s^{TT}_4)$ $\exists$ $(s^{TS}_2\ a^{TS}_2\ s^{TS}_3)$.

6. The TTICU requests an update of a policy that has not been enforced and the TSICU does not acknowledge the policy update: for $\forall$ $(s^{TT}_5\ a^{TT}_5\ s^{TT}_6)$, $(s^{TT}_6\ a^{TT}_7\ s^{TT}_5)$ $\exists$ $(s^{TS}_3\ a^{TS}_2\ s^{TS}_3)$.

7. The TTICU deletes an enforced policy: for $\forall$ $(s^{TT}_4\ a^{TT}_8\ s^{TT}_7)$, $(s^{TT}_7\ a^{TT}_9\ s^{TT}_8)$ $\exists$ $(s^{TS}_2\ a^{TS}_3\ s^{TS}_4)$.

8. The TTICU deletes a policy that is not enforced: for $\forall$ $(s^{TT}_5\ a^{TT}_8\ s^{TT}_7)$, $(s^{TT}_7\ a^{TT}_9\ s^{TT}_8)$ $\exists$ $(s^{TS}_3\ a^{TS}_3\ s^{TS}_4)$.

Therefore, $T_{TT}$ and $T_{TS}$ have a weak bi-simulation relationship, i.e., they expose equivalent behaviors. □

In addition to policy management, the R1 interface also provides railway enrichment information services. The RMAO gathers information from ETCS entities and from external information sources. Using this information, the TTICU can derive information that can help the internal functions or applications of both the TTICU and the TSICU. The purpose of R1 enrichment information is to enable the TSICU to improve the optimization of the railway operation performance by utilizing information that is not available in the railway network. The information sources can be derived by external sources and provided by the TTICU via the R1 interface. There may exist different types of enrichment information, and by identifying the enrichment information type, the TSICU may request the delivery of such information. The application programming interfaces of the enrichment information service may also follow the REST principles.

## 6. An Experiment and Discussions

To emulate the R1 policy management service, a numerical experiment is set up. In particular, the experiment is aimed at evaluating the latency in provisioning feedback policy data from different TSICUs to a TTICU.

The RESTful approach that is adopted here imposes the well-known client–server pattern, and the setup for the numerical experiment is shown in Figure 13.

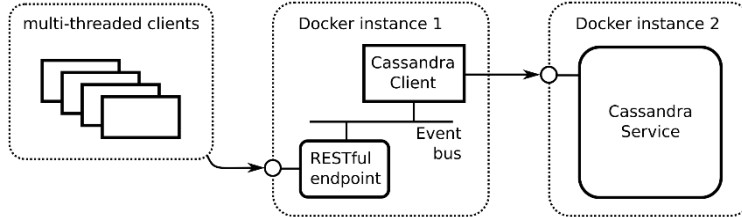

**Figure 13.** Setup of the numerical experiment—traffic from TSICUs to a TTICU.

The RESTful endpoint (EP) that is exposed by the policy management service is implemented using Vertx [37]. This toolkit has proven properties, such as multi-language support, flexibility etc., and aside from that, it includes an Apache Cassandra [38] client

in order to facilitate the integration. Moreover, a core component within the toolkit is the event bus, which makes internal message exchange possible in a very resource-efficient and asynchronous way.

The choice of Apache Cassandra as a NoSql distributed database backend is based on its maturity, high availability, and scalability. The service virtualization as whole is achieved by the use of Docker [39] container instances for the lightness and robustness that it shows.

The RESTful service EP is used over HTTP/TCP/IPv6/GbE, where the interface and the IPv6 addresses are isolated in order to allow the experiment to be as unaffected by any other traffic as possible. The clients access the operations of the service in a multi-threaded way, and the clients' traffic is consisted of notification operations, i.e., POST requests, such as the request/response pattern shown in Figure 10.

The notification operation shown in Figure 10 consists of a request, which includes the EP's URI, headers such as Host, Content-type, Accept, etc., plus an experimental header to keep track of the moment that the request is created (it is subsequently copied into the response as it is received by the EP) in a local location for the respective client nano-time scale, and the JSON description of the operation placed in the request body part.

The traffic, that is generated for latency estimation purposes, is of a volume of twenty thousand operations of the notification type, i.e., request/response pairs.

Thus, the recorded latency for each operation is stored in nanoseconds, where the latencies of the fifth and 15th time frames of one thousand operation are shown in Figures 14 and 15, respectively.

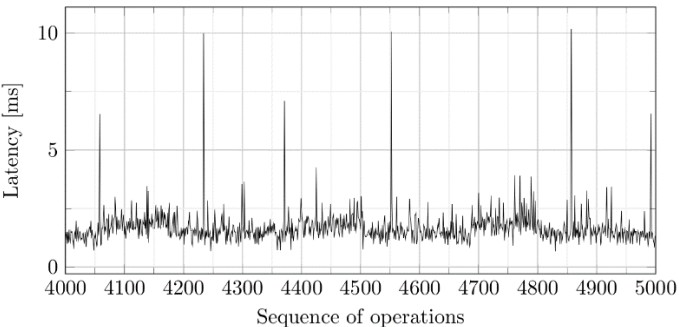

**Figure 14.** Sequence of latency values measured within the fifth group of one thousand operations.

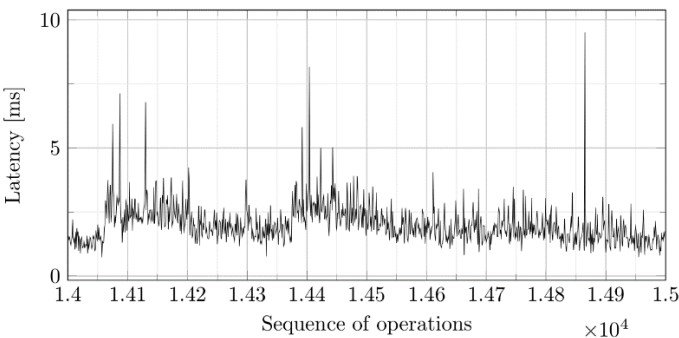

**Figure 15.** Sequence of latency values measured within the 15th group of one thousand operations.

The sequence of latency values might be seen as a stochastic process; thus, a typical tool for depicting such a process is the creation of the respective probability density functions, as shown in Figure 16.

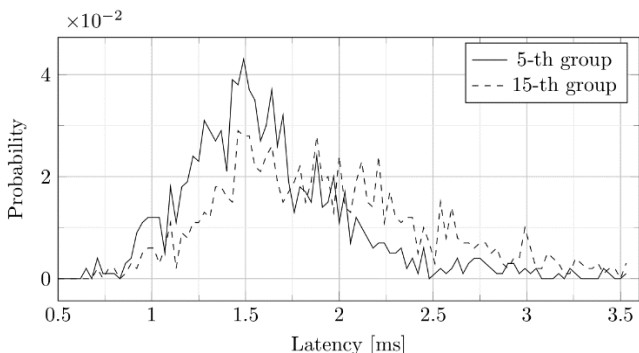

**Figure 16.** Probability density functions for latency values measured in the fifth and 15th groups of one thousand operations.

Despite of the observable similarity of the curves in their shapes, one must rather conclude that the process is quite far from the definition for a stationary one.

The average latency values, which are shown in Table 1, are within the limit of 2–4 ms when averaged over the whole time frame, but it is easy to notice that the top 5% of latency values gave a higher average than that of the lower 95%; in some cases, the values were up to almost 10 times worse.

**Table 1.** Average latency values in milliseconds for the time frames.

| Time Frame Number | 0–95% of Latency Samples | 95–100% of Latency Samples | 0–100% of Latency Samples |
|---|---|---|---|
| 5th | 1.381 | 3.702 | 1.613 |
| 15th | 2.222 | 19.516 | 3.951 |

## 7. Conclusions

The paper presents a new intelligent railway network architecture that applies the principles of disaggregation, programmability, and openness. The proposed architecture embeds intelligence in the RBC's functionality, which is disaggregated into time-tolerant functions and time-sensitive functions. The interaction between time-tolerant and time-sensitive services is defined in an application programming interface that follows the REST principles. This programmability can improve efficiency, reliability, and flexibility, and it simplifies the implementation of railway functions.

The proposed intelligent railway network architecture supports open interfaces between logical functions that can be implemented on general-purpose hardware. It also allows software and hardware from different vendors for the TTICU and TSICU. The disaggregation and open interfaces between the decoupled railway network components provide efficient interoperability between multiple vendors. Another major principle of the proposed intelligent architecture is the virtualization of railway network functions, which reduces costs.

This paper illustrates the role of identified logical components of an intelligent railway network with a use case. Some details of the application programming interface between time-tolerant and time-sensitive services are discussed. The latency over the interface, as one of its key performance indicators, was evaluated with an emulation.

**Author Contributions:** Conceptualization, E.P. and V.T.; methodology, E.P.; software, I.A.; validation, I.A. All authors have read and agreed to the published version of the manuscript.

**Funding:** This research was funded by the Bulgarian National Science Fund under Grant No. KP-06-H37/33.

**Conflicts of Interest:** The authors declare no conflict of interest.

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
