# Peer review of "Towards Intelligent, Programmable, and Open Railway Networks"

_applsci, doi:10.3390/app12084062_

Round 1

Reviewer 1 Report

The article was written in a very interesting and appropriate language. In addition, it contains properly described abstract keywords and the title corresponds to the content. In the introduction, the authors defined the research problem, described the structure of the article, and referred to the research problem in the context of the research task, even putting forward a hypothesis. The methodical and empirical part of the article raises no objections. The article has a coherent and logical structure, the conclusions are sufficient. The authors referred to the rich and up-to-date literature.

The only drawback is not adapted to the pattern at the end (in some cases) - the list of references.

Nevertheless, the authors made a great positive impression on me in terms of the balance of the threads and their mature approach to the subject.

Author Response

The authors are grateful for valuable comments.

Comment: The only drawback is not adapted to the pattern at the end (in some cases) - the list of references.

Answer: In the revised version, all references are formatted as to the journal template.

English language spell check and miner format changes are done.

Reviewer 2 Report

This paper proposed an architecture of railway network function virtualization. It focuses on the time-sensitive and time-tolerant services and the interaction between them. The motivation are explained in a clear way. The function and the architecture of ETCS have been analyzed, especially the future RBC architecture evolution related to disaggregation is highlighted, which would be the core application to achieve an intelligent, programmable and open railway networks.

However, although the authors specified the interface between TTICU and TSICU in detail in chapter 5, it has not been explained how exact the horizontal and vertical disaggregation of RBC could be realized in the architecture. In addition, there are some small remarks:

Page 4 Row 128, 131, 132 RRC should be changed as RBC

The term AI, ML and AI/ML are not used in a consistent way. It is not clear if they have different contexts of use, or just mixed used.

The explain on IR interface is insufficient. Suggest to add more explanation or remove it.

Author Response

The authors are grateful for valuable comments.

Comment 1: Although the authors specified the interface between TTICU and TSICU in detail in chapter 5, it has not been explained how exact the horizontal and vertical disaggregation of RBC could be realized in the architecture.

Answer: The following paragraph is inserted to explain the horizontal and vertical disaggregation: “Horizontal disaggregation can be achieved by the separation of RBC time-tolerant functions from RBC time-sensitive functions using open interfaces. The separation of RBC functionality aims at running of different control loops with distinct latency requirements. The interface openness can be achieved through application programming interfaces (APIs) which facilitate interoperability. The valuable benefits of API are related with low cost and easy implementation, more flexible functionality, cloud-based deployment, etc. Vertical disaggregation decouples hardware and software by applying RBC function virtualization. Function virtualization can be achieved by transition from software running on proprietary and specialized hardware to software running on general purpose hardware. Among the advantages of function virtualization are reduction of maintenance costs, easier function upgrades, scalability, greater agility, and flexibility.”

Comment 2: Page 4 Row 128, 131, 132 RRC should be changed as RBC

Answer: In the revised version, changes have been made.

Comment 3: The term AI, ML and AI/ML are not used in a consistent way. It is not clear if they have different contexts of use, or just mixed used.

Answer: The following paragraph is inserted: “Currently, the AI technologies mostly used in railways include ML, Robotics and Natural Language Processing [21]. In the rest of the paper, the focus is on ML in the railway transport, used to perform tasks, or to make predictions, based on experience or example data. In this context, ML can be used for sensing and predicting. For example, sensing can be useful in image recognition of dangerous situations, image analysis for obstacle detection, fault detection and for maintenance of tunnels, bridges, buildings etc. Predicting is useful in e. g. in preventive maintenance of the equipment, and can improve railway safety, punctuality, reliability, and availability.”

AI/ML is substituted by ML, after the above explanation.

Comment 4: The explain on IR interface is insufficient. Suggest to add more explanation or remove it.

Answer: In the revised version, the following paragraph is added: “The procedures executed over the IR interface are related to exchange of information between ETCS entities, or notifications about internal events or events triggered by external entities and the required reaction. These procedures define the required status and mode changes of the ETCS entities (train driver, the trackside equipment, the on-board equipment) on a context level. Examples of procedures include mission starting and mission ending, shunting, overriding e. g. in case of failed signal, train joining or splitting, train orientation changing, indicating of track conditions. The full list of procedures and their detailed description may be found in [31].”

English language spell check and miner format changes are done.